# Differences in Overweight or Obesity, Changes in Dietary Habits after Studying Abroad and Sleep Quality by Acculturative Degree among Asian Foreign Students: A Cross Sectional Pilot Study

**DOI:** 10.3390/ijerph19095370

**Published:** 2022-04-28

**Authors:** Miae Doo, Chunyang Wang

**Affiliations:** Department of Food and Nutrition, Kunsan National University, Daehak-ro 558, Gunsan 54150, Korea; yangyangyang0302@gmail.com

**Keywords:** acculturative degree, dietary habits, foreign students, obesity, overweight, sleep pattern

## Abstract

The number of foreign students is increasing worldwide, and they suffer from acculturation to different environments or cultures. This pilot study examined the difference in overweight or obesity, changes in dietary habits after studying abroad and sleep quality according to acculturative degree among 225 Asian foreign students in South Korea. Most subjects (61.8%) experienced a low acculturative degree. The change in dietary habits after studying abroad showed a significant difference according to the acculturative degree (*p* < 0.001); however, there were no differences observed in sleep quality (*p* = 0.090) and prevalence of overweight or obesity according to acculturative degree (*p* = 0.101). Interestingly, a difference in the risk for being overweight or obese by sleep quality after being stratified into groups according to acculturative degree was observed. Among the groups with a low acculturative degree, subjects reporting poor sleep quality had a 2.875-fold (95% CI = 1.167–7.080) higher risk of being overweight or obese than those reporting good sleep quality. However, the risk of being overweight or obese was not different among the high acculturative group regardless of their sleep quality. The results showed that the degree of acculturation could influence the risk of being overweight or obese according to sleep quality among Asian foreign students.

## 1. Introduction

The number of foreign students registered in South Korean higher education institutions has gradually increased over the past decades. According to data from the Ministry of Education for South Korea, the number of foreign students was 89.54 thousand in 2011 but increased to 153.70 thousand in 2020 [1]. Almost 91.9% were Asian foreign students among the foreign students, and in descending order, were mainly from China, Vietnam, Uzbekistan, and Mongolia.

Foreign students who come to study face various difficulties in environmental and cultural differences from their original countries [2,3]. These processes are defined as acculturation in adopting new environmental and cultural aspects, such as another language, climate, attitudes, beliefs and different dietary cultures [3,4,5]. The acculturative degree in foreign students could affect health outcomes as well as academic achievement [2,6]. For example, foreign students with a poor acculturative degree (those defined as high acculturative stress) show poor sleep, appetite disturbance, psychological distress, health consequences, and academic problems [2,3,4,5,6].

Obesity, which is considered to be the main risk factor for various chronic diseases, is influenced by various factors, such as diet, physical activity, socioeconomic factors, sleep patterns, and genetic factors [7]. Dietary habits are well known to be important factors in the prevalence and management of obesity [8,9]. Epidemiological studies have reported that the risk and prevalence of overweight and obesity have increased, while sleep duration has decreased over the past half-century [10]. Additionally, many researchers have consistently suggested that a worse sleep pattern, including shorter duration or poor quality of sleep, is associated with overweight or obesity [11,12,13].

Although the associations of sleep quality or dietary habits with obesity are well known, studies on the difference in sleep quality or dietary habits according to the acculturative degree and the relationship between these differences and obesity among foreign students are rare. We hypothesized that foreign students suffer changes in sleep quality or dietary habits after studying abroad. These changes, which differ depending on the acculturative degree, were related to overweight or obesity. Therefore, the aim of this pilot study was to examine the difference in changes in dietary habits after studying abroad or sleep patterns according to acculturative degree. Additionally, we determined whether these differences affected the acculturative degree with the risk of being overweight or obese among Asian foreign students studying in South Korea.

## 2. Materials and Methods

### 2.1. Subjects and Collection

Our study was approved by the Institutional Review Board of the Kunsan National University (IRB No. 1040117-201910-HR-018-03) before collecting the study subjects. With the help of the Institute of International Exchange and Language Education, subjects were recruited through the organization unit or club for foreign students. Study subjects were collected among Asian foreign students at universities in Jeonbook, South Korea, from October 2019 to January 2020. At the time of participation in the survey, subjects were included who (1) were of a foreign nationality, (2) currently resided in South Korea, and (3) who had studied in South Korea for at least a semester. Subjects voluntarily participated, and before initiating the questionnaire, a written informed consent form was obtained. This study included a total of 233 Asian foreign students (112 men and 121 women).

The survey was carried out in an enclosed space for 25–30 min. All bilingual interviewers conducted face-to-face interviews. At the beginning of the interview, questions were asked in Korean, and then the interview was conducted in Korean, Chinese, or English according to the subject’s preference.

### 2.2. Measures and Data Collection

This study was carried out using a newly developed questionnaire. The questionnaires were composed of data on demographic characteristics, health-related measurements, sleep patterns, anthropometric variables, and changes in dietary habits after studying abroad to examine the relationship among changes in dietary habits after studying abroad and sleep quality by acculturative degree among Asian foreign students in South Korea. Because the questionnaire was developed in Korean, it was translated into Chinese, English, and Vietnamese versions for foreign students. The translation of the questionnaire was supervised by professors or experts in each language.

The acculturative degree was assessed based on the degree of adapting to Korean culture and skills [14]. It consists of eight questions scored on a 5-point Likert scale, with a score of 1 indicating “strongly disagree” and 5 indicating “strongly agree”. Higher scores indicated a high acculturative degree. Subjects were divided into “LA (low acculturated degree; <median value 115.00 score)” or “HA (high acculturated degree; ≥median value 115.00 score)” using a cutoff score of median value. The Cronbach’s alpha values of a previous study [14] and our study were 0.73 and 0.72, respectively.

For demographic characteristics, the subjects’ age, gender, period of stay in South Korea, academic course, and Korean conversational ability were collected. Health-related variables, such as current smoking, alcohol consumption, subjective health status, and acculturative stress, were assessed. Subjects were divided into “drinkers” or “nondrinkers” based on their current alcohol consumption status and “smokers” and “nonsmokers” based on their current smoking status. Their subjective health status was assessed using a 4-point Likert scale, and higher scores indicated a higher subjective health status. Acculturative stress levels were assessed using the acculturative stress scale for international students [15]. It consists of 36 questions with seven subscales, including “perceived discrimination”, “homesickness”, “perceived hate”, “fear”, “culture shock”, “guilt”, and “miscellaneous”. The responses of subjects were scored on a 5-point Likert scale, with a score of 1 indicating “strongly disagree” and 5 indicating “strongly agree”. The overall scores possible on this scale ranged from 35 to 180, with higher scores indicating a higher perceived stress status from acculturation. Several previous studies reported Cronbach’s alpha values of 0.93 or above [16,17], and the Cronbach’s alpha of this study was 0.89.

Subjects’ body weight and height, as anthropometric variables, were assessed using a self-reported method. Overweight and obesity are determined by body mass index (BMI), which is calculated using body weight and height body weight in kilograms divided by height in meters squared (BMI = body weight [kg]/height [m^2^]). According to the Asia-Pacific Regional Office of the World Health Organization, being overweight or obese is defined as having over 23.0 kg/m^2^ [18].

The changes in dietary habits scale was used to assess the changes in the dietary habits of foreign students after studying in South Korea from their county in accordance with our previous study [19]. It is composed of three subscales across 10 questions about changes in the environment of dietary consumption, 10 questions about dietary problems, and five questions about reasons for unbalanced dietary habits after studying abroad. The questions were scored on a 5-point Likert scale, with a score of 0 indicating “strongly disagree”, 2.5 indicating “disagree”, 5 indicating “neutral”, 7.5 indicating “agree” and 10 indicating “strongly agree”. Higher scores indicate poor dietary habits. The Cronbach’s alpha was 0.74.

As usual, sleep quality during the previous month was assessed by using the Pittsburgh Sleep Quality Index (PSQI) [20]. It comprised seven subscales on “perceived sleep quality”, “sleep latency”, “sleep duration”, “sleep efficiency”, “sleep disturbances”, “use of sleeping medication”, and “daytime dysfunction”. Higher scores indicate poor quality sleep. The subjects were categorized into “good sleepers (PSQI ≤ 5)” or “poor sleepers (PSQI > 5)” using a cutoff score of 5.

### 2.3. Statistical Analyses

All collected data were analyzed using SPSS (version 27.0, IBM Corp., Armonk, NY, USA) software for Windows. Generalized linear models were examined to identify differences in the general characteristics, changes in eating habits after studying abroad and sleep quality according to acculturative degree after adjustment for age and gender. The odds ratio for being overweight or obese according to acculturative degree and various factors was examined using multinomial logistic regression models after adjusting for age and gender. These various factors were considered when assessing sleep quality and changes in dietary habits after studying abroad. After dividing by the acculturative degree scores, multinomial logistic regression models were used to determine how the odds ratio for being overweight or obese was modified by sleep quality [good sleep quality (PSQI ≤ 5) as a reference] or changes in dietary habits after studying abroad [good dietary habits (≥median value 115.00 score) as a reference]. Statistical significance was considered at a *p* value of <0.05.

## 3. Results

The general characteristics of subjects stratified by acculturative degree are presented in Table 1. The average age and proportions of men were 26.58 years and 48.1%, respectively. Additionally, more than half of the subjects (61.8%) had a low acculturative degree. Subjects in the high acculturative degree group had a significantly longer period of stay in South Korea (*p* = 0.001), were more often graduate students (*p* < 0.001), and had a better Korean conversational ability (*p* < 0.001) than those in the low acculturative degree group. Acculturative stress was significantly different between the HA and LA groups (*p* = 0.017). Additionally, perceived discrimination (*p* = 0.010) and perceived hate (*p* = 0.005) among acculturative stress were significantly different in accordance with acculturative degree. However, there were no differences in subjective health status (*p* = 0.190), smoking (*p* = 0.980), alcohol consumption (*p* = 0.316), BMI (*p* = 0.052) or prevalence of overweight and obesity (*p* = 0.101) by acculturative degree.

The changes in dietary habits after studying abroad were divided according to acculturative degree and are represented in Table 2. The low acculturative degree group showed a significantly higher total score of change in dietary habits after studying abroad than the high acculturative degree group (118.16 ± 27.47 vs. 101.16 ± 26.67, *p* < 0.001). Additionally, subscales among the change in dietary habits after studying abroad, the dietary problem and the reason for unbalanced dietary habits different significantly by acculturative degree (41.20 ± 16.39 vs. 34.07 ± 15.85, *p* = 0.001 for the dietary problems and 22.62 ± 9.45 vs. 16.43 ± 6.44, *p* < 0.001 for the reason for unbalanced dietary habits), with the exception of change in the environment of dietary consumption (*p* = 0.058). Among the changes in the environment for dietary consumption, the subjects with a low acculturative degree showed higher scores for ‘The range of food options has been reduced’. (6.11 ± 2.64 vs. 4.24 ± 3.29, *p* < 0.001), ‘There are difficulties in communicating when buying food’. (3.77 ± 2.59 vs. 2.64 ± 2.57, *p* = 0.001), and ‘I can’t eat what I like’. (6.16 ± 2.96 vs. 3.93 ± 2.77, *p* < 0.001) than those with a high acculturative degree, whereas lower scores for ‘I started to eat mainly Korean food.’ (6.89 ± 2.51 vs. 7.75 ± 2.56, *p* = 0.011) and ‘There is an economic problem.’ (6.61 ± 2.57 vs. 7.36 ± 2.83, *p* = 0.036). Among the dietary problems, ‘I can’t eat well because I don’t have time.’ (3.68 ± 3.18 vs. 2.08 ± 2.67, *p* < 0.001), ‘I only eat when I’m hungry’ (4.01 ± 3.37 vs. 2.81 ± 2.87, *p* = 0.006), ‘The number of binge dietary times depending on the taste of the food has increased’. (5.77 ± 3.13 vs. 3.82 ± 3.02, *p* < 0.001), ‘The number of times breakfast is skipped to binge lunch has increased.’ (3.51 ± 3.06 vs. 2.36 ± 2.59, *p* = 0.004), and ‘My meat intake has increased’. (4.53 ± 3.01 vs. 3.01 ± 3.20, *p* <0.001) scored significantly higher in the LA group, but ‘I purchase and eat fast food often’. (5.28 ± 3.28 vs. 6.18 ± 3.18, *p* = 0.029) scored lower than in the HA group. Among the reasons for unbalanced dietary habits, all subgroups significantly different between the LA and HA groups. In other words, the low acculturative degree group showed higher scores for ‘I think this is because I can buy food easily in my home country, but it is difficult to buy food in Korea’. (5.17 ± 2.74 vs. 2.75 ± 2.44, *p* < 0.001), ‘I think it’s because Korea doesn’t have access to food that I eat every day in my home country’. (4.17 ± 2.76 vs. 2.13 ± 2.08, *p* < 0.001), ‘I think this is because I can cook and eat food in my home country, but I can’t eat the same foods in Korea.’ (4.90 ± 3.50 vs. 3.15 ± 2.81, *p* < 0.001), and ‘I think it’s mainly because I eat out’. (4.24 ± 2.96 vs. 3.09 ± 2.58, *p* = 0.003), whereas lower scores for ‘I think it is because I eat alone when I am in Korea, and I eat with family at home’. (4.15 ± 3.02 vs. 5. 31 ± 2.87, *p* = 0.004).

The sleep pattern, including sleep duration and quality assessed by the PSQI, is shown by acculturative degree in Table 3. There were no significant differences in the global PSQI score (*p* = 0.090) or sleep duration (*p* = 0.838). However, among the PSQI components, sleep latency (1.22 ± 0.87 vs. 0.97 ± 0.68, *p* = 0.012) and daytime dysfunction (1.28 ± 0.80 vs. 1.07 ± 0.70, *p* = 0.033) were significantly different between the LA and HA groups.

Although a difference in the prevalence of overweight and obesity according to acculturative degree was not observed (Table 1), the effect of sleep quality or dietary habits after studying abroad on the risk of being overweight or obese after being stratified into acculturative degree groups was identified using multinomial logistic regression after adjusting for age and gender (Figure 1). Among the subjects in the low acculturative group, poor sleepers had a 2.875 times (95% CI = 1.167–7.080) higher risk for being overweight and obese than good sleepers (Figure 1A). In contrast, no significant difference in the risk for being overweight or obese was revealed by sleep quality among the subjects in the high acculturative group. Additionally, after being stratified into groups according to dietary habits, there were no observed differences in the risk for being overweight or obese between the low and high acculturative groups (Figure 1B).

**Table 2 ijerph-19-05370-t002:** Changes in dietary habits after studying abroad according to acculturative degree.

Changes in Dietary Habits	Total (*n* = 233)	HA (*n* = 89)	LA (*n* = 144)	*p* Value *
I. Change in environment of dietary consumption	52.90 ± 14.73	50.56 ± 15.10	54.34 ± 14.36	0.058
The number of food purchases has increased.	5.34 ± 3.47	5.34 ± 3.87	5.35 ± 3.22	0.974
The number of cooked meals has increased.	3.42 ± 3.55	3.29 ± 3.87	3.51 ± 3.35	0.669
The range of food options has been reduced.	5.40 ± 3.04	4.24 ± 3.29	6.11 ± 2.64	<0.001
There is no one who helps make food.	5.97 ± 3.46	5.84 ± 3.59	6.04 ± 3.38	0.731
The time to eat has decreased.	4.49 ± 2.91	4.16 ± 3.15	4.69 ± 2.74	0.177
There are difficulties in communicating when buying food.	3.34 ± 2.64	2.64 ± 2.57	3.77 ± 2.59	0.001
I can’t eat what I like.	5.31 ± 3.08	3.93 ± 2.77	6.16 ± 2.96	<0.001
I started to eat mainly Korean food.	7.22 ± 2.56	7.75 ± 2.56	6.89 ± 2.51	0.011
There is an economic problem.	6.90 ± 2.69	7.36 ± 2.83	6.61 ± 2.57	0.036
When choosing food, the influence of a friend is great.	5.52 ± 3.44	6.01 ± 3.26	5.21 ± 3.53	0.070
II. Dietary problems	38.48 ± 16.52	34.07 ± 15.85	41.20 ± 16.39	0.001
The irregular hours of work and rest lead to irregular mealtimes.	5.43 ± 3.27	5.42 ± 3.42	5.43 ± 3.19	0.971
I can’t eat well because I don’t have time.	3.07 ± 3.09	2.08 ± 2.67	3.68 ± 3.18	<0.001
I only eat when I’m hungry.	3.55 ± 3.23	2.81 ± 2.87	4.01 ± 3.37	0.006
The number of binge dietary times depending on the taste of the food has increased.	5.02 ± 3.22	3.82 ± 3.02	5.77 ± 3.13	<0.001
The number of meals skipped to lose weight has increased.	2.65 ± 3.08	2.67 ± 3.28	2.64 ± 2.95	0.847
The number of times breakfast is skipped to binge lunch has increased.	3.07 ± 2.76	2.36 ± 2.59	3.51 ± 3.06	0.004
I tend to drink heavily.	2.21 ± 2.76	2.28 ± 2.73	2.17 ± 2.78	0.864
My vegetable intake has decreased.	3.91 ± 3.00	3.46 ± 3.05	4.18 ± 2.94	0.065
My meat intake has increased.	3.95 ± 3.17	3.01 ± 3.20	4.53 ± 3.01	<0.001
I purchase and eat fast food often.	5.62 ± 3.26	6.18 ± 3.18	5.28 ± 3.28	0.029
III. Reason for an unbalanced dietary habits	20.26 ± 8.94	16.43 ± 6.44	22.62 ± 9.45	<0.001
I think it is because I eat alone when I am in Korea, and I eat with family at home.	4.59 ± 3.01	5.31 ± 2.87	4.15 ± 3.02	0.004
I think this is because I can buy food easily in my home country, but it is difficult to buy food in Korea.	4.25 ± 2.88	2.75 ± 2.44	5.17 ± 2.74	<0.001
I think it’s because Korea doesn’t have access to food that I eat every day in my home country.	3.39 ± 2.70	2.13 ± 2.08	4.17 ± 2.76	<0.001
I think this is because I can cook and eat food in my home country, but I can’t eat the same foods in Korea.	4.23 ± 3.35	3.15 ± 2.81	4.90 ± 3.50	<0.001
I think it’s mainly because I eat out.	3.80 ± 2.87	3.09 ± 2.58	4.24 ± 2.96	0.003
Total score of dietary habits	111.63 ± 28.36	101.07 ± 26.67	118.16 ± 27.47	<0.001

Values represent means ± SD. * *p* values were calculated using a generalized linear model adjusted for age and gender. HA, High acculturative degree (acculturative degree ≥ 3); LA, Low acculturative degree (acculturative degree < 3).

**Table 3 ijerph-19-05370-t003:** Sleep pattern according to acculturative degree.

	Total (*n* = 233)	HA (*n* = 89)	LA (*n* = 144)	*p* Value *
Components of PSQI				
1. Sleep quality	1.18 ± 0.65	1.12 ± 0.54	1.22 ± 0.71	0.296
2. Sleep latency	1.12 ± 0.81	0.97 ± 0.68	1.22 ± 0.87	0.012
3. Sleep duration	0.79 ± 0.84	0.83 ± 0.80	0.77 ± 0.86	0.574
4. Habitual sleep efficiency	0.45 ± 0.76	0.40 ± 0.64	0.48 ± 0.83	0.419
5. Sleep disturbances	0.96 ± 0.54	0.93 ± 0.58	0.98 ± 0.51	0.510
6. Sleeping medication	0.19 ± 0.51	0.21 ± 0.49	0.17 ± 0.52	0.667
7. Daytime dysfunction	1.20 ± 0.77	1.07 ± 0.70	1.28 ± 0.80	0.033
Global PSQI score	5.90 ± 2.73	5.54 ± 2.57	6.12 ± 2.82	0.090
Sleep duration (hours)	6.61 ± 1.12	6.59 ± 0.89	6.62 ± 1.25	0.838

Values represent means ± SD. * *p* values were calculated using a generalized linear model adjusted for age and gender. HA, High acculturative degree (acculturative degree ≥ 3); LA, Low acculturative degree (acculturative degree < 3).

**Figure 1 ijerph-19-05370-f001:**
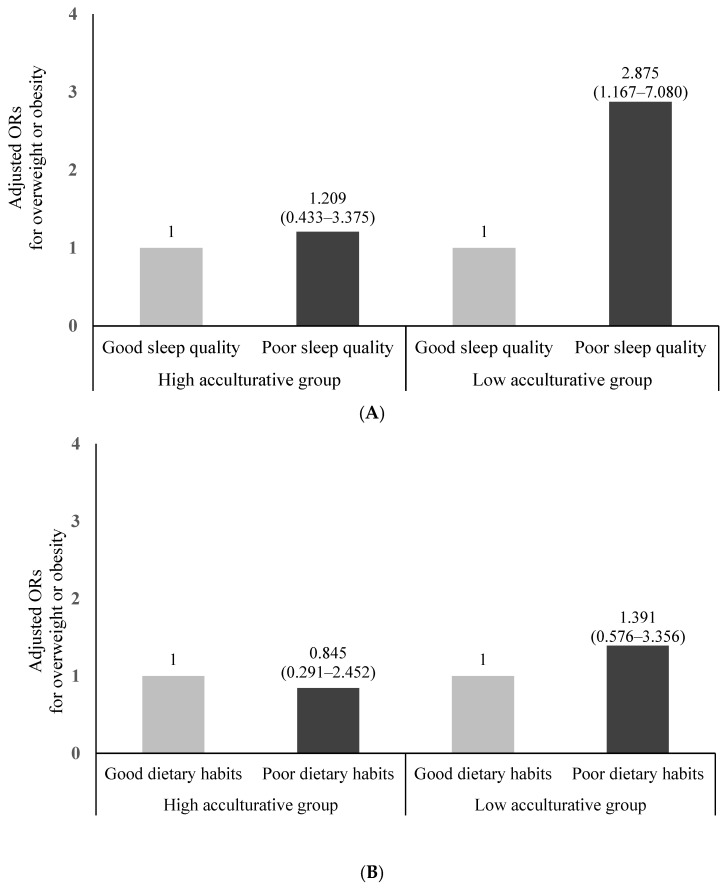
The adjusted odds ratios (ORs) and 95% confidence intervals (95% CIs) for overweight and obesity according to acculturative degree and various factors. (**A**) Stratified by sleep quality [good sleep quality (PSQI ≤ 5) as a reference]; (**B**) stratified by changes in dietary habits after studying abroad [good dietary habits (≥median value 115.00 score) as a reference]. The adjusted odds ratios were calculated using multinomial logistic regression for age and gender.

## 4. Discussion

Among foreign Asian students who come to South Korea to study, 61.8% of students who participated in this study reported a low acculturative degree. An acculturative degree in foreign students was associated with their period of stay in South Korea, prevalence of graduate students, Korean conversational ability, acculturative stress and change in dietary habits after studying abroad. Additionally, when divided by acculturative degree, a significant difference was observed in the risk of being overweight or obese according to sleep quality using multinomial logistic regression after adjusting for age and gender; among the group with a low acculturative degree, the subjects reporting poor sleep quality showed an increase in the risk for being overweight or obese, whereas the group with a good acculturative degree did not differ in the risk of being overweight or obese regardless of their sleep quality.

Foreign students who showed a high acculturative degree had a longer period of residence in South Korea, were graduate students, fluent in Korean and reported low acculturative stress, which is consistent with the findings of previous studies [21,22,23]. Generally, foreign students with a higher conversational ability in the host country easily socialize and assimilate the host cultures [21,22], and a longer length of residence in the host country has many opportunities to experience the host language, cultures, and society, which means that it is well adopted in a new environment (host country) [23]. Specifically, foreign students with limited host-country-language proficiency have increased academic difficulties due to language barriers and suffer significant levels of acculturative stress [2].

Evidence from previous studies indicate that among immigrants as well as foreign students, there is an association between acculturative degree and the influence on health behaviors [3,6,24]. Of the various health behaviors of foreign students, dietary habits, in particular, may be greatly affected by acculturative degree [5,6,25], which is in agreement with this study. In this study, subjects with a high acculturative degree showed a more positive influence of changes in dietary habits after studying abroad than those with a low acculturative degree. However, other studies conducted in Europe and North America have suggested that people who are well acculturated tend to have a negative effect of dietary habits [6,25,26]. Negative changes in dietary habits are high in fat, leads to the consumption of fewer fresh fruits and vegetables and a high frequency of consumption of fast food or processed snack foods, which explains Western dietary acculturation from an individual’s host country. The positive findings of our study could be explained by similar dietary cultures, including low fat, high carbohydrate, and plant-based meals [27,28].

Differences in overweight or obesity according to acculturative degree were not observed in this study. These results are inconsistent with previous studies suggesting an increased body mass index with more acculturation [25,26]. However, a combined effect of sleep quality on an individual’s acculturative degree was observed with respect to the risk of being overweight or obese. The risk of being overweight or obese among poor sleepers, especially among the group with a low acculturative degree, increased compared with those who were good sleepers. Conversely, such findings were not observed among the group with a high acculturative degree regardless of their sleep quality. Although it is difficult to ascertain the direction of the causal association between the risk for being overweight or obese, sleep quality and acculturative degree, these findings could be explained by a number of other studies [2,3,4,5,6,25,26]. A poor acculturative degree is associated with increased psychological distress or lower sleep quality, which is caused by psychological problems. Generally, psychological or sleep problems may lead to decreased dietary habits, which is a risk factor for overweight or obesity. Therefore, these findings suggested that adopting a new culture (acculturative degree) could be maintained by improving sleep quality through programs that provide psychological support, which could be used as a strategy to promote academic achievement and assist foreign students in building their health status.

This pilot study demonstrated that a low acculturative degree in relation to poor quality sleep could lead to the increased risk of being overweight or obese. However, these results should be interpreted carefully because of the major limitation of its nature as a cross-sectional study. Another limitation of our study is that the subjects were Asian, and the sample size was small. Therefore, as a pilot study, it is difficult to generalize these findings to foreign students in other countries, and further studies using these findings are needed to conduct studies with the same design and large sample sizes.

## 5. Conclusions

As a cross-sectional pilot study of Asian foreign students, this study demonstrated that acculturative degree was positively related with dietary habits, which was inconsistent with studies carried out in Western countries. Interestingly, acculturative degree was associated with increase in the risk for being overweight or obese in relation to sleep quality. However, these findings from our study are necessary to validate more clear results with large diverse foreign student population.

## Figures and Tables

**Table 1 ijerph-19-05370-t001:** General characteristics according to acculturative degree.

General Characteristics	Total (*n* = 233)	HA (*n* = 89)	LA (*n* = 144)	*p*-Value *
Gender (men)	48.1	50.6	46.5	-
Age (years)	26.58 ± 5.22	26.58 ± 5.22	26.29 ± 5.80	-
Period of stay in South Korea (months)	25.05 ± 0.97	29.79 ± 17.59	22.13 ± 12.02	0.001
Academic course (≥graduate school course)	58.4	70.8	55.7	<0.001
Korean conversational ability (≥middle class)	27.9	44.9	17.4	<0.001
Subjective health status	2.03 ± 0.62	1.96 ± 0.50	2.07 ± 0.69	0.190
Acculturative stress	84.36 ± 16.90	80.94 ± 15.14	86.48 ± 17.63	0.017
Perceived discrimination	16.94 ± 4.64	15.93 ± 3.42	17.56 ± 5.17	0.010
Homesickness	10.80 ± 2.85	10.47 ± 2.63	11.01 ± 2.97	0.174
Perceived hate	10.59 ± 3.37	9.81 ± 2.74	11.08 ± 3.64	0.005
Fear	9.16 ± 2.52	8.91 ± 2.81	9.31 ± 2.76	0.268
Culture shock	7.89 ± 2.24	7.74 ± 2.34	7.99 ± 2.17	0.431
Guilt	3.89 ± 1.57	3.67 ± 1.40	4.03 ± 1.66	0.067
Miscellaneous	25.08 ± 5.54	24.40 ± 5.12	25.50 ± 5.77	0.167
Smoking (smoker)	20.6	21.3	20.1	0.980
Alcohol consumption (drinker)	79.4	83.1	77.1	0.316
BMI (kg/m^2^)	23.52 ± 7.65	24.85 ± 8.54	22.71 ± 6.95	0.052
Prevalence of overweight and obesity	25.3	31.5	21.5	0.101

Values represent the means ± SD or % (*n*). * *p* values were calculated using a generalized linear model adjusted for age and gender. HA, High acculturative degree (acculturative degree ≥ 3); LA, Low acculturative degree (acculturative degree < 3).

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
