# Peer review of "Differences in Overweight or Obesity, Changes in Dietary Habits after Studying Abroad and Sleep Quality by Acculturative Degree among Asian Foreign Students: A Cross Sectional Pilot Study"

_ijerph, 2022, doi:10.3390/ijerph19095370_

Round 1

Reviewer 1 Report

Thank you for the opportunity to review this paper. This study mainly examined how the effect of acculturative degree on the risk of being overweight and obesity is modified via sleep quality or dietary habits after studying abroad among Asian foreign students studying abroad in South Korea. The topic is original and relevant in this field, since the acculturation is a great part of life for immigrants, which has been shown to be associated with their health status. A lot of acculturation studies in western countries found negative health effects due to acculturation, and this study added contribution to literature since it thinks about this topic from a different perspective. It is a well-written paper, the conclusions are consistent with the evidence and arguments presented and the references are appropriate, but I have a few comments to improve this manuscript.

Lines 53-56: based on your method and results, I think you have more study aims than simply measure the moderation effects.

Lines 56-66: it is better to provide more detailed information about how you recruited the sample.

Lines 68-69: “face-to-face interviews” sounds ambiguous, did you read questions to participants? More details about study procedure are needed. 

Lines 77-81: is the scales for acculturative degree designed by yourself or adapted from existing scales? Did you validate the scales other than reporting the Cronbach's alpha?

Measures:

Be clearer that you averaged some subscales or scales into index before running your regressions.

Figures:

Be clearer about what the y-axis indicates?

Lines 251-253: this sentence is incomplete, please rewrite.

Lines 248-251 are repeated with Lines 217-212.

Lines 239-241: I am glad you identified these studies and pointed out the difference in results between those studies and your study.

Lines 264-265: I am glad you mentioned the limitation of small sample size. With this being side, I suggest you reframe your study as a pilot study and reflect this change in your title. 

Author Response

Thank you for the opportunity to review this paper. This study mainly examined how the effect of acculturative degree on the risk of being overweight and obesity is modified via sleep quality or dietary habits after studying abroad among Asian foreign students studying abroad in South Korea. The topic is original and relevant in this field, since the acculturation is a great part of life for immigrants, which has been shown to be associated with their health status. A lot of acculturation studies in western countries found negative health effects due to acculturation, and this study added contribution to literature since it thinks about this topic from a different perspective. It is a well-written paper, the conclusions are consistent with the evidence and arguments presented and the references are appropriate, but I have a few comments to improve this manuscript.

We appreciate the reviewer for the careful reading and description of our manuscript with the valuable comments. We worked to the best of my abilities to revise the issues the reviewer pointed out.

Lines 53-56: based on your method and results, I think you have more study aims than simply measure the moderation effects.

It has been revised based on the comments as follows: line 53-57.

“Therefore, the aim of this pilot study was to examine the difference in change in dietary habits after studying abroad or sleep pattern according to acculturative degree. Additionally, we determined whether these differences affected acculturative degree on the risk of being overweight or obese among Asian foreign students studying abroad in South Korea.

Lines 56-66: it is better to provide more detailed information about how you recruited the sample.

Thank you for your comment.

We have rewritten the information for recruiting the subjects as follows: 62-68.

“With the help of the Institute of International Exchange and Language Education, subjects were recruited through the organization unit or club for foreign students. Study subjects were collected among Asian foreign students at universities in Jeonbook, South Korea, from October 2019 to January 2020. At the time of participation in the survey, subjects were included to (1) have a foreign nationality, (2) currently reside in South Korea, and (3) study in South Korea for at least a semester.”

Lines 68-69: “face-to-face interviews” sounds ambiguous, did you read questions to participants? More details about study procedure are needed.

As you suggested, the interview procedure has been rewritten as follows: 71-74.

“The survey was carried out in an enclosed space for 25-30 minutes. All bilingual interviewers conducted the form of face-to-face interviews. At the beginning of the interview, questions were asked in Korean, and then interview was conducted in Korean, Chinese, or English according to the subject’s preference.”

Lines 77-81: is the scales for acculturative degree designed by yourself or adapted from existing scales? Did you validate the scales other than reporting the Cronbach's alpha?

We have added a reference and a previous study of Cronbach's alpha.

“The acculturative degree was assessed based on the degree of adapting to Korean culture and skills [14].~

The Cronbach's alpha values of a previous study [14] and our study were 0.73 and 0.72, respectively.”

Measures: Be clearer that you averaged some subscales or scales into index before running your regressions.

It has been modified on the comments as follows: line 87-90 and line 124-125.

“Subjects were divided into “LA (low acculturated degree; < median value 115.00 score)” or “HA (high acculturated degree; ≥ median value 115.00 score)” using a cutoff score of median value.”

“The subjects were categorized into “good sleepers (PSQI ≤5)” or “poor sleepers (PSQI >5)” using a cutoff score of 5.”

Figures: Be clearer about what the y-axis indicates?

It has been modified on the comments

Lines 251-253: this sentence is incomplete, please rewrite.

Thank you for your comment.

It has been rewritten as your comment as follows: line 26--265

“Although it is difficult to ascertain the direction of the causal association between risk for being overweight or obese, sleep quality and acculturative degree, these findings could be explained by the many studies. “

Lines 248-251 are repeated with Lines 217-212.

As you pointed out, we agree that paragraphs 1 and 4 appear to be repeated. However, paragraph 1 summarizes the overall content of this study, and paragraph 4 discusses the combined effect of sleep quality on acculturative degree with respect to the risk of being overweight or obese. Therefore, we worded the sentence differently.

Lines 239-241: I am glad you identified these studies and pointed out the difference in results between those studies and your study.

Based on your comment, we has been discussed as follow: line 252-254.

The positive findings of our study could be explained by similar dietary cultures, including low fat, high carbohydrate, and plant-based meals [27, 28].

Lines 264-265: I am glad you mentioned the limitation of small sample size. With this being side, I suggest you reframe your study as a pilot study and reflect this change in your title.

Thank you for your comments.

Based on your comment, we have been reframed as a pilot study and changed the title of our study.

“Differences in overweight or obesity, changes in dietary habits after studying abroad and sleep quality by acculturative degree among Asian foreign students: A cross-sectional pilot study”

In the DISCISSION:

“ This pilot study demonstrated that a low acculturative degree when they slept at poor quality could influence the increased risk of being overweight or obese.”

“ Therefore, as a pilot study, it is difficult to generalize these findings to foreign students in other countries, and further studies using these findings are needed to conduct studies with the same design and large population sample sizes”

Reviewer 2 Report

This work appears to be novel.

You discuss the accultative and acculturative stress assessments but you do not indicate if these assessments have been validated.  

While I appreciate your use of the validated, Sleep Quality Index, however, I would like to have seen that the health status assessments were validated.  If it was, I think that you need to include more details of your assessment.  

You indicate greater acculturation among students older in age and in graduate school.  I think including both undergraduate and graduate students may skew the results.  I would encourage you to evaluate undergraduates and graduate student separately

In the discussion, paragraph four, you indicate that “overweight or obesity according to acculturative degree were not observed in this study”, however, in the next paragraph you indicate “This study demonstrated that low acculturative degree could influence the increased risk of being overweight or obesity…”. I think these two conclusions are contradictory.

Author Response

This work appears to be novel.

We appreciate the reviewer for the careful reading and description of our manuscript with the valuable comments.

You discuss the accultative and acculturative stress assessments but you do not indicate if these assessments have been validated.

Thank you for your comments.

We have added a reference and presented Cronbach's alpha of previous studies as follows: line 89-90, line 104-105.

“ The Cronbach's alpha values of a previous study [14] and our study were 0.73 and 0.72, respectively.”

“Several previous studies reported Cronbach's alpha values of 0.93 or above [16, 17], and the Cronbach's alpha of this study was 0.89.”

  1. Wang, J.; Kang, Y.E.; Lee, S.Y. Stress and Dietary Behavior by Acculturation Level among Chinese Students Living in Korea. J East Asian Soc Diet Life. 2019, 29, 42-55.

  1. Yu, B.; Chen, X.; Li, S.; Liu, Y.; Jacques-Tiura, A.J.; Yan, H. Acculturative stress and influential factors among international students in China: a structural dynamic perspective. PLoS One. 2014, 9, e96322.

  1. Iorga M, Soponaru C, Muraru ID, Socolov S, Petrariu FD. Factors Associated with Acculturative Stress among International Medical Students. Biomed Res Int. 2020;2020:2564725.

While I appreciate your use of the validated, Sleep Quality Index, however, I would like to have seen that the health status assessments were validated. If it was, I think that you need to include more details of your assessment.

Thank you for the comment.

There is only one question related to subjective health status with a 4-point Likert scale, with a score of 1 indicating “unhealthy” and 4 indicating “very healthy.” Therefore, it is not appropriate to assess validation.

You indicate greater acculturation among students older in age and in graduate school. I think including both undergraduate and graduate students may skew the results. I would encourage you to evaluate undergraduates and graduate student separately.

We really appreciate the constructive and very helpful comments. Like your comment, we agreed that academic course (undergraduate or graduate) could influence acculturation. Our results confirm that graduate students have a higher acculturative degree. However, as mentioned in the limitations of the DISCUSSION, the sample size of our study was too small, so when divided by academic course, it is difficult to confirm the tendency. However, age, which could affect the results, was statistically treated as a covariate.

In the discussion, paragraph four, you indicate that “overweight or obesity according to acculturative degree were not observed in this study”, however, in the next paragraph you indicate “This study demonstrated that low acculturative degree could influence the increased risk of being overweight or obesity…”. I think these two conclusions are contradictory.

Thank you for the comment.

It has been revised based on the comments as follows: line 272-273.

“ This pilot study demonstrated that a low acculturative degree when they slept at poor quality could influence the increased risk of being overweight or obese.”

Reviewer 3 Report

Specific and general comments are listed below:

  1. For the most part I liked the paper and the use of the English language was adequate. But there are some spots where the use of language needs to be improved.
  2. line 36 - use a word rather than "problems."
  3. line 54 - in several spots the word "obesity" is used when the word "obese" is correct.
  4. line 56 - delete "abroad."
  5. line 63 - these dates cannot be correct.
  6. line 68 - "examine"  use new word
  7. line 74 - delete "of the questionnaire.."
  8. line 95 - it seems that the number should be 36
  9. line 107 - If the scale went from 0-10 it seems like it would be more than a 5-point scale.
  10. line 111 - "usual"
  11. Stat analysis - you write about the "effect" of things and the "risk".  Your study is cross-sectional and these words should be replaced. 
  12. line 168 - "purchase"
  13. Discussion - how do you suggest that people improve their sleep quality?
  14. line 265 - delete "too"

Author Response

Specific and general comments are listed below:

For the most part I liked the paper and the use of the English language was adequate. But there are some spots where the use of language needs to be improved.

We appreciate the reviewer for the careful reading and description of our manuscript with the valuable comments. We worked to the best of my abilities to revise the issues the reviewer pointed out.

line 36 - use a word rather than "problems."

It has been modified based on the comments.

line 54 - in several spots the word "obesity" is used when the word "obese" is correct.

Based on your comments, it has been revised throughout the entire manuscript.

line 56 - delete "abroad."

It has been deleted based on the comments.

line 63 - these dates cannot be correct.

It has been modified based on the comments.

line 68 - "examine" use new word

It has been modified based on the comments.

line 74 - delete "of the questionnaire."

It has been deleted based on the comments.

line 95 - it seems that the number should be 36

It has been deleted.

line 107 - If the scale went from 0-10 it seems like it would be more than a 5-point scale.

It has been rewritten as follows: line 116-119.

 “The questions were scored on a 5-point Likert scale, with a score of 0 indicating “strongly disagree”, 2.5 indicating “disagree”, 5 indicating “neutral”, 7.5 indicating “agree” and 10 indicating “strongly agree.”

line 111 - "usual"

It has been modified based on the comments.

Stat analysis - you write about the "effect" of things and the "risk". Your study is cross-sectional and these words should be replaced.

Based on your comments, it has been revised in statistical analyses as follows: line 131-138.

The odds ratio for being overweight or obese according to acculturative degree and various factors was examined using multinomial logistic regression models after adjusting for age and gender. These various factors were considered sleep quality and changes in dietary habits after studying abroad. After dividing by acculturative degree scores, multinomial logistic regression models were used to determine how the odds ratio for being overweight or obese was modified by sleep quality [good sleep quality (PSQI ≤5) as a reference] or changes in dietary habits after studying abroad [good dietary habits (≥ median value 115.00 score) as a reference].

line 168 - "purchase"

It has been modified based on the comments.

Discussion - how do you suggest that people improve their sleep quality?

We consider psychological factors to be highly related to sleep. Therefore, it has been presented in DISCUSSION as follows: line 268-271.

“Therefore, these findings suggested that adopting a new culture (well acculturative degree) could be maintained by improving sleep quality through programs such as psychological support, which could be used as a strategy to promote academic achievement and assist foreign students in building health status.”

line 265 - delete "too"

It has been deleted based on the comments.

Round 2

Reviewer 2 Report

I appreciate your edits.   I feel that these updates have improved the paper greatly.  I believe these edits have provided a more clear indication of the data and the implications of this research